# Production of Dense Cu-10Sn Part by Laser Powder Bed Fusion with Low Surface Roughness and High Dimensional Accuracy

**DOI:** 10.3390/ma15093352

**Published:** 2022-05-07

**Authors:** Flaviana Calignano, Diego Manfredi, Silvia Marola, Mariangela Lombardi, Luca Iuliano

**Affiliations:** 1Department of Management and Production Engineering (DIGEP), Politecnico di Torino, Corso Duca Degli Abruzzi, 24, 10129 Torino, Italy; flaviana.calignano@polito.it (F.C.); luca.iuliano@polito.it (L.I.); 2Integrated Additive Manufacturing Center (IAM)—Politecnico di Torino, Corso Castelfidardo, 51, 10129 Torino, Italy; mariangela.lombardi@polito.it; 3Department of Applied Science and Technology (DISAT), Politecnico di Torino, Corso Duca Degli Abruzzi, 24, 10129 Torino, Italy; 4Center for Sustainable Future Technologies, Istituto Italiano di Tecnologia (IIT), Via Livorno 60, 10144 Torino, Italy; silvia.marola@iit.it

**Keywords:** tin-bronze alloys, laser powder bed fusion, Archimedes’ density, optical microscopy, surface roughness, volumetric energy density

## Abstract

Tin-bronze alloys with a tin content of at least 10 wt% have excellent mechanical properties, wear resistance, and corrosion resistance. Among these alloys, Cu-10Sn was investigated in this study for production with the laser powder bed fusion process with a 500W Yb:YAG laser. In particular, a design of experiment (DoE) was developed in order to identify the optimal process parameters to obtain full density, low surface roughness, and high dimensional accuracy. Samples were characterized with Archimedes’ method and optical microscopy to determine their final density. It was shown that the first method is fast but not as reliable as the second one. A first mechanical characterization was performed through microhardness tests. Finally, a set of process parameters was identified to produce fully dense samples with low surface roughness and high accuracy. The results showed that the volumetric energy density could represent an approach that is too simplified, therefore limiting the direct correlation with the physical aspects of the process.

## 1. Introduction

The continuous evolution of additive manufacturing (AM) processes for metals is continuing to attract increasing attention from sectors in which there is the widespread use of copper alloys. Among the additive technologies, the laser powder bed fusion (L-PBF) process, also known as selective laser melting (SLM), mainly attracts the attention of the sectors that need a dense product with a good dimensional accuracy of the components [1]. The high reflectivity of Cu at the wavelength of the lasers commonly used in the majority of L-PBF commercial systems, combined with its high thermal conductivity, causes a certain instability in the process, as well as the risk of damaging the optical mirror of the L-PBF machine due to the sustained copper back-reflections [2]. For this reason, although there are new studies on lasers with different wavelengths, many researchers have investigated the use of copper alloys with lower optical reflectivity than pure copper. Considering the literature on materials, although constant research is being conducted for the development of new high-performance copper alloys for the L-PBF process [3,4,5,6,7], bronze, one of the oldest metal alloys known to mankind, is still widely used in many industrial applications. The fields of application range from works of art with bronze sculptures and musical instruments, to applications of technological importance such as electrical connectors and high precision springs. Furthermore, bronze, thanks to its excellent resistance to salt-water corrosion, is widely used in marine applications for propulsion systems and seawater treatment [8]. Generally, when the tin content represents at least 10% of its weight, bronze has good mechanical properties, corrosion resistance, and wear resistance. With respect to the majority of metallic alloys with L-PBF, Cu-10Sn bronze does not require post-process heat treatment [9,10] which would increase the process time and costs and, therefore, the properties depend on the microstructure generated during solidification. Since bronze L-PBF components can potentially be used without post-processing, it is of fundamental importance that the process parameters employed to produce the components are optimized in such a way as to guarantee the obtainment of dense and accurate parts with low surface roughness. The close link between these properties and the process parameters has been widely documented in the literature, e.g., [11]. Tan et al. [12] investigated the Cu-10Sn alloy with a body-center-cubic (BCC) lattice structure produced using an EOS M280 machine equipped with a laser power of 400 W. The search for optimal process parameters is carried out by giving great importance to the numerical value of the volumetric energy density and analyzing its effects on density. Their analyses show that by using an energy density of 84.7 J/mm^3^ with a scan speed of 1100 mm/s and a hatching distance of 0.09 mm, a microstructure with few micro-cracks and pores is obtained. Using a lower or higher energy density induces the formation of pores. Another study by Scudino et al. [10] compared the microstructure and mechanical properties of the specimens produced in Cu-10Sn using L-PBF technology with those obtained by casting. The specimens were produced using an SLM^®^ 250 HL device equipped with a Yb-YAG laser and the following process parameters were used: scan speed of 210 mm/s, laser power of 271 W, layer thickness of 90 µm and hatching distance of 90 µm. Even though this study was not focused on process-parameter optimization, the analysis highlighted a certain strengthening of the material accompanied by a notable improvement in ductility due to the fine microstructure typical of the L-PBF process, demonstrating the effectiveness of this technology for the production of materials with improved mechanical performance. Deng et al. [13] also produced Cu–10Sn alloys with a 3D printer EP-M100 T machine at three different energy densities and reported an optimum energy density of 220 J/mm^3^. They used the following process parameters: laser power of 95 W, layer thickness of 0.02 mm, hatching distance of 0.06 mm and three energy density levels, specifically 210, 220 and 230 J/mm^3^ to investigate the trend of density and mechanical properties when varying the laser energy density. They found that a linear relationship of these properties with the laser energy density does not exist and that the best properties were achieved at 220 J/mm^3^. Zeng et al. [14] investigated the compositions, microstructures, and the mechanical, thermal, and corrosion properties of Cu-10Sn specimens produced with a Concept-Laser Mlab cusing R system. The specimens, produced with a laser with a power of 95 W, scanning speed of 1200 mm/s, hatching distance of 50 μm, and layer thickness of 15 μm, were analyzed in the as-fabricated (AF) condition and after vacuum annealing (VA) at 600 ℃ and 800 ℃. AF samples exhibit smaller grain sizes and higher compressive strength than those of VA samples. The thermal conductivity of the AF samples is higher than that of VA samples. However, the corrosion rate of AF samples is higher than that of VA samples. Mehta et al. [15], based on the process parameters analyzed, and using an SLM^®^ 125 HL machine equipped with a 400 W Yb fiber laser, highlighted that extremely low scanning speeds allow for dense samples with superior tensile strength and ductility in comparison to the conventionally fabricated alloy. The value ranges of the analyzed process parameters were between 200 and 350 W for laser power and 100 and 1000 mm/s for scanning speed. The hatching distance and layer thickness were kept constant at 0.12 mm and 0.03 mm, respectively. Nevertheless, in all these works, no explorations of surface roughness and dimensional accuracy of components were carried out.

In the present study, the objective was precisely to obtain optimized process parameters that guarantee full density, low surface roughness, and good dimensional accuracy of the Cu-10Sn parts taking into account productivity. This aspect, coupled with a high density, is highly important when considering the additive process as a suitable technology for the industrial production of components in this alloy. Therefore, a design of experiment (DoE) study on the main process parameters was first conducted to identify a process window for dense parts with low roughness. The density of the samples was measured with Archimedes’ method and optical microscopy. Then, the results of microhardness, surface roughness and accuracy were correlated to the productivity achievable with the L-PBF system. At the end, the necessary parameters for obtaining components during its final application are outlined.

## 2. Materials and Methods

### 2.1. Powder

The feedstock material employed for the production of the samples was a gas atomized tin bronze powder (90% Cu and 10% Sn) provided by EOS GmbH. Powder characterization was performed following the ASTM F3049-14 standard. A Field Emission Scanning Electron Microscope (FESEM), ZEISS SUPRA TM 40 (Jena, Germany), was used to observe the morphology of the feedstock powder and of the as-built components. The raw powder was spherical with minor satellites, as shown in Figure 1. The particle size distribution was measured using the Mastersizer 3000 laser diffraction particle size analyzer from Malvern Panalytical (Malvern, UK). The particle size distribution was observed to be monomodal with d_10_, d_50_ and d_90_ at 16 μm, 28 μm and 44 μm, respectively.

The composition of the alloy was checked after the L-PBF process using an Optical Emission Spectrometer Metal Lab Plus S7 of GNR srl Analytical Instruments Group (Novara, Italy), and it is reported in Table 1. No important deviations from the composition of the powder declared by the producer were evidenced.

### 2.2. L-PBF Processing

To produce the L-PBF test specimens, a Print Sharp 250 machine (Prima Additive, Torino, Italy) equipped with a 500 W Yb:YAG laser (wavelength of 1060–1080 nm) was used. A stainless-steel building platform set at 80 °C was employed as the substrate for the production of the test specimens [12,16]. Platform preheating was employed mainly to avoid moisture content in the powders and to help reduce the formation of residual stresses in the components during the process. The processing chamber was flooded with inert gas (argon) to keep the oxygen content below 100 ppm (0.01%) during the processing period. To identify the process parameters for manufacturing Cu-10Sn parts with a high density, the Taguchi method was used. Taguchi’s designs are generally very fragmented so as to greatly reduce the cost and time required for a designed experiment. The main parameters investigated were laser power *P*, scan speed *v*, and hatching distance *h_d_* [17]. Test specimens of 15 × 15 × 10 mm^3^ in size were produced using a scanning strategy with a 67 degree rotation of each exposed layer with respect to the previous one in order to reduce thermal stresses during the overlapping of the various layers [18].

The Taguchi method is based on orthogonal array experiments which provide reduced variance for the experiment with optimum settings for the control parameters. Signal-to-Noise ratios (S/N) are the objective functions used to analyze data and they are calculated in the case of larger-the-better or smaller-the-better problems, using Equation (1) or Equation (2), respectively:*η_L_* = −10 *log*_10_ [(1/*n*) Σ (1/*y_i_*^2^)]     [dB](1)
*η_S_* = −10 *log*_10_ [(1/*n*) Σ *y_i_*^2^]        [dB] (2)
where *η_L_* (or *η_S_*) indicates the S/N ratio calculated from the observed values, *n* represents the number of repetitions of each experiment, and *y_i_* is the experimentally observed value of the *i*-th experiment. Smaller-the-better and larger-the better problems were used for the roughness and density, respectively. An *L*_18_ orthogonal array with 17 degrees of freedom (Table 2) was used to reduce the number of tests from 54, which is the number of tests used in traditional full-factorial experimental plans, to 18. With an analysis of variance (ANOVA) we calculated the statistical confidence.

Before developing the DoE, a preliminary test campaign was conducted to determine the range of parameter values that would facilitate at least the production of the samples. For inadequate values, a delamination of the first layers and consequent interruption of the additive process was observed. The first set of the preliminary campaign was formulated based on the value of the volumetric energy density (*E_d_ = P/v* × *h_d_* × *t*) [6] and by making an approximate comparison with the value of the melting enthalpy of the material used. The choice of the values for the DoE reported in Table 2 was also based on the calculation of the volume building rate, also called the material build rate [19] and build-up rate [20]. A high-volume building rate can be obtained with a high scan speed, a larger hatching distance, or/and large layer thickness, thus reducing the processing time.

### 2.3. Characterization

A variety of methods have been proposed in the literature for density evaluations of parts produced by AM processes. They can be non-destructive or destructive methods. In the first case, the most used are computed tomography, which is limited to the instrument resolution, and Archimedes’ method (ASTM B962-17). In the second case, the samples are cut and polished and then analyzed usually with optical microscopy (OM) [21]. Spierings et al. [21] suggested that Archimedes’ method is economical, fast, and reliable for the quantification of L-PBF samples porosity. However, in a more recent study, du Plessis et al. [22] highlighted the problems associated with the Archimedes’ method. Inclusions, for example, could increase the measured mass and thus change the measured density and the assumed nominal density for alloys with varying compositions.

For this reason, in the present study, all the samples manufactured with all the sets of parameters derived from the DoE were characterized using Archimedes’ method and optical microscopy (OM) to investigate their density and, conversely, their porosity. In the Archimedes method, each sample is weighed three times in air and then in water, using the appropriate balance. With OM, the density of the samples was determined by image analysis using ImageJ software on at least 20 micrographs per sample at a 50× magnification.

Vickers microhardness measurements (ASTM E384-17) were performed on samples polished up to 3 µm, through a Future Tech FM-810 Micro Vickers hardness tester (Assago, Italy), and by applying a load of 500 gf and a dwell time of 15 s. A mean HV value of five measurements was obtained for all samples. 

Considering surface roughness, in terms of values of average roughness (R_a_) and average maximum height of the profile (R_z_), the samples were characterized in as-built conditions with the use of an SM Metrology Systems RPT80 tester (Volpiano, Italy). The measurement distance was 4.8 mm and a 0.8 mm cut-off filter was used (ISO 4288:1997). Five measurements were made for each surface and the arithmetic mean of the measurements was used for the analysis. Images of the as-built sample surfaces were obtained with an FESEM analysis. After the roughness of the as-built samples was measured, the samples were subjected to shot blasting to remove any incompletely melted powder. New roughness measurements were then carried out and the values were compared. The Norblast SD9 shot blasting machine (Bologna, Italy) was used with a glass microsphere and a pressure of 6 bar.

The dimensional accuracy of the samples was also analyzed. The samples were scanned using a structured light 3D scanner, ATOS Compact Scan 2M from GOM (Buccinasco, Italy), and were compared with the CAD model using inspection software (GOM Inspect Professional) to obtain indications for geometric deviations.

## 3. Results and Discussion

### 3.1. Density

All specimens demonstrated an average density of 99.6 ± 0.3% considering that the nominal density of this alloy is 8.78 g/cm^3^ [9] (Figure 2a and Table 3).

During the OM analysis, the samples were found to be dense with at least very fine isolated spherical pores (Figure 2b), while with Archimedes’ method, values higher than the theoretical density value were obtained. Considering that a density of higher than 100% is meaningless in Table 3, only the experimental density obtained from the OM analysis was reported. This finding confirms that this latest method could be fast and simple, but it is not reliable when detecting porosity differences for samples with a density higher than 99%. Density measurements with OM are certainly more time consuming, but also more precise, as is well documented in the foundry and casting literature.

The analyzed process parameters have an energy density ranging from 64.4 J/mm^3^ to 236.6 J/mm^3^. Regardless of the energy value, all the samples examined under the microscope did not show the presence of cracks contrary to what was reported by Tan et al. [12] for the Cu-10Sn alloy and showed full density. The energy values for density therefore differ from those reported in the literature. The volumetric energy density is often criticized [17,23] because the absorbance of the laser energy throughout the layer is dependent on various parameters such as the packing density of the material, and it assumes the penetration depth of the laser complies with the thickness of the produced layer, which is generally not the case. Moreover, because of the Gaussian temperature distribution of the laser beam, the heat introduced into the powder bed is not homogenous throughout the laser diameter, since the highest temperature would occur at the innermost region. Thus, the particles being hit by the center of the laser beam are exposed to higher temperatures than the ones at the edges. Depending on the diameter of the laser beam and on the hatching distance, there is an overlap of the single scan paths of the moving heat source, and therefore, some points are exposed multiple times. If the beam moves with an increased scan speed, the overlapping area decreases because of the reduced time for which the powder is exposed to the laser. As a result of these effects, it becomes clear that the equations are simplified and limit direct correlation with the real process. Table 4 and Figure 3 show the influence of scan speed, laser power and hatching distance on the relative density with OM.

According to a range analysis from S/N (Table 4), the significance order of process parameters for density is laser power > scan speed > hatching distance. By analyzing the data shown in Figure 3, it is possible to see that the trend of the curves is similar for all parameters except for the scan speed of 800 mm/s. When the lines are parallel, it means that there is no interaction. The more the lines deviate from parallelism, the greater the degree of interaction. It is clear that the differences between the data are very low. For this reason, the choice of parameters for the construction of the samples could be made by considering those parameters that lead to greater productivity. Looking at the volume rate (Table 3), these parameters are a scan speed of 800 mm/s, laser power of 170 W and hatching distance of 0.11 mm.

Samples produced with these parameters have a mean hardness of 163 ± 2 HV (Figure 4 and Table 3). This hardness value is among the three highest values but corresponds to the lowest density value (98.70%). The highest hardness values are 166 HV and 165 HV which correspond to densities of 99.4 and 99.7 and a productivity value of 5.36 cm^3^/h and 3.35 cm^3^/h, respectively. Therefore, at an industrial level, such a low difference in values necessarily leads to the choice of parameters that lead to enhanced production (9.50 cm^3^/h). 

### 3.2. Surface Roughness

The surface roughness values in the upper part of the samples as-built (Figure 5) is equal to 7.21 µm ± 2.03 µm and 35.60 µm ± 10.16 µm for *R_a_* and *R_z_*, respectively.

Table 5 shows the results of the ANOVA for the response variable *R_a_.* The ANOVA is used to determine which factor has the most significant effect with respect to all controllable factors on the roughness of the samples produced by L-PBF. From the ANOVA, laser power was found to have the most significant effect on surface roughness. A linear model analysis shows the coefficients for each factor, their *p*-values and the ANOVA table. All interaction terms and factors are significant at an α-level of 0.10 for S/N ratios.

Based on the analysis of the range from S/N (Table 6), the significance order of process parameters on surface roughness is first the laser power, then the scan speed, and finally the hatching distance.

Through an interaction graph (Figure 6), it is possible to visualize the effects between the variables. An increase in laser power from 170 to 195 W led to a reduction in roughness for all scan speeds except 720 mm/s, where there an increase in roughness was observed. In that section, the scan speeds of 500 mm/s, 680 mm/s and 800 mm/s had a greater eventuality, highlighting a greater reduction in the roughness compared to the other scan speeds.

Considering the effect of the hatching distance, its increase from 0.05 mm to 0.08 mm led to a reduction in roughness by about half for a speed of 500 mm/s. A less pronounced reduction was observed for a scan speed of 680 mm/s and 800 mm/s. For these last two scan speeds, the effect of the hatching distance is evident, from 0.08 mm to 0.11 mm. Considering instead the interaction between the laser power and the hatching distance, for a laser power of 220 W there is a different trend than the other two laser powers. The parameters that obtain the least roughness are a laser power of 220 W, scan speed of 560 mm/s and hatching distance of 0.11 mm (Figure 6). 

The roughness values on the contour are *R_a_* = 10.81 μm and *R_z_* = 66.76 μm. After shot blasting with glass beads, the values of roughness on the contour decreased to *R_a_* = 3.13 μm and *R_z_* = 16.42 μm. This reduction is due to the removal of the not completely melted particles which adhere to the surface due to the heat on the powder bed (Figure 7 and Figure 8). Through shot blasting it is possible to reduce the roughness as required to reach the specifications of the standards.

### 3.3. Accuracy

The data on the accuracy in the x- and y-directions (Figure 9) show a certain uniformity in both directions. By analysing Figure 10, it is possible to see that as scan speed increases, the samples increase their accuracy. In these cases, for the same scan speed, the effect of laser power and hatching distance is, for the values chosen, irrelevant for accuracy. On the contrary, the choice of laser power and hatching distance affects accuracy for scanning speed values of lower than 680 mm/s.

## 4. Conclusions

Cu-10Sn samples were fabricated with the additive L-PBF process using a 500 W Yb:YAG laser and the effect of the process parameters on density, microhardness, surface roughness and accuracy was examined. A design of experiment based on the Taguchi method was employed to reduce the number of samples from 54 to 18. The main conclusions are as follows:The parameters that can help to achieve a dense structure with good productivity and high dimensional accuracy are as follows: a scan speed of 800 mm/s, laser power of 170 W, hatching distance of 0.11 mm and layer thickness of 30 μm.The parameters that can help to achieve a good surface roughness are as follows: laser power of 220 W, scan speed of 560 mm/s and hatching distance of 0.11 mm.The volumetric energy density does not provide a correct analysis of the energy necessary to melt the material to obtain dense components without cracks and pores due to its numerical expression which is excessively simplified with respect to the process. This could lead to the indication of energy values which can provide incorrect information.The determination of the density of samples produced with L-PBF using the Archimedes’ method is fast, but it is not reliable if the density of samples exceeds 99%. To evaluate in a more precise way the porosity of dense samples, careful image analysis should be carried out.

## Figures and Tables

**Figure 1 materials-15-03352-f001:**
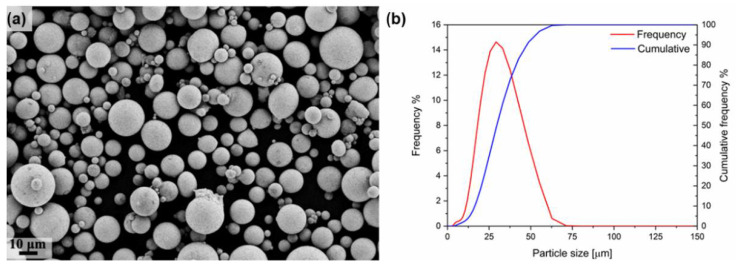
(**a**) Morphology of the tin bronze powder. (**b**) Particle size distribution of the tin bronze powder.

**Figure 2 materials-15-03352-f002:**
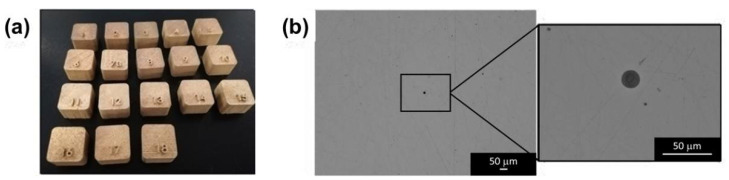
(**a**) Picture of the Cu-10Sn samples produced by LPBF with the DoE. (**b**) OM images of Cu-10Sn samples for density measurements at lower (**left**) and higher (**right**) magnification. Only a very small spherical pore could be detected.

**Figure 3 materials-15-03352-f003:**
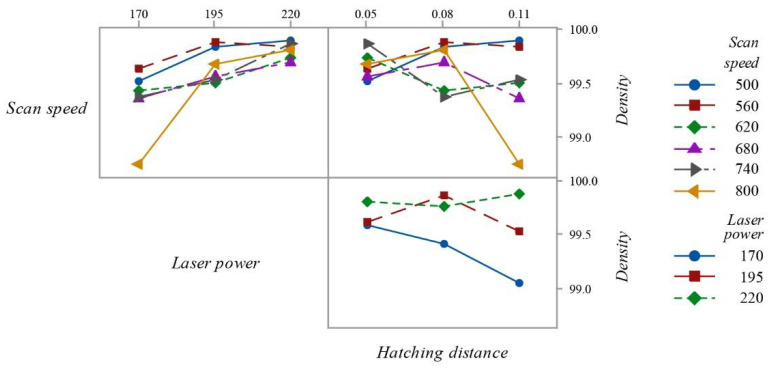
Interaction plot for density.

**Figure 4 materials-15-03352-f004:**
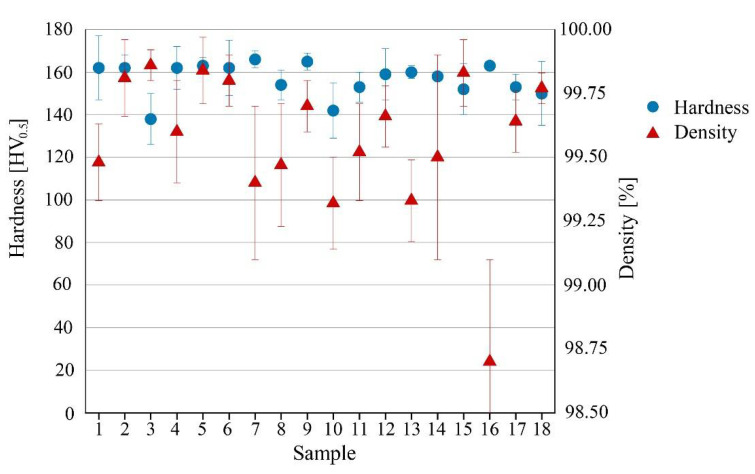
Micro Vickers hardness and density of the 18 samples under analysis.

**Figure 5 materials-15-03352-f005:**
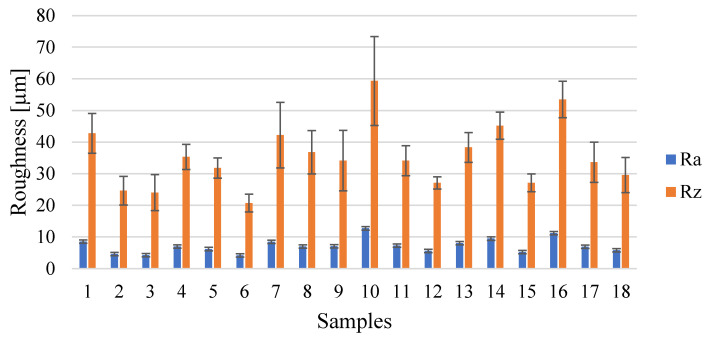
Results of *R_a_* [µm] and *R_z_* [µm].

**Figure 6 materials-15-03352-f006:**
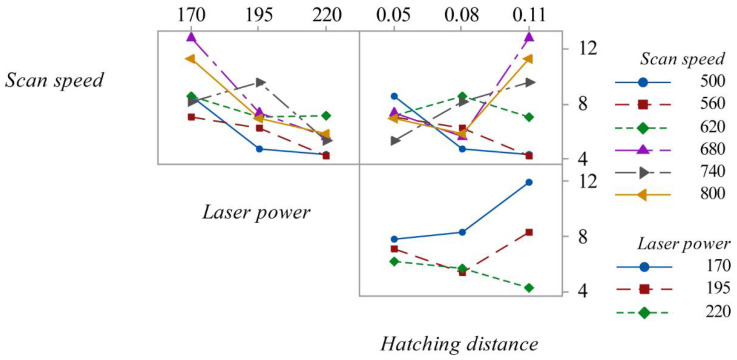
Interaction plot for *R_a_*.

**Figure 7 materials-15-03352-f007:**
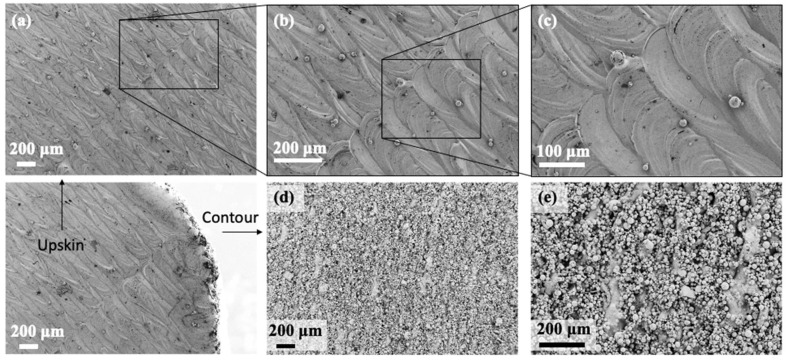
FESEM images of surface roughness of the sample 6 as-built: (**a**–**c**) on top surface; (**d**,**e**) lateral surface.

**Figure 8 materials-15-03352-f008:**
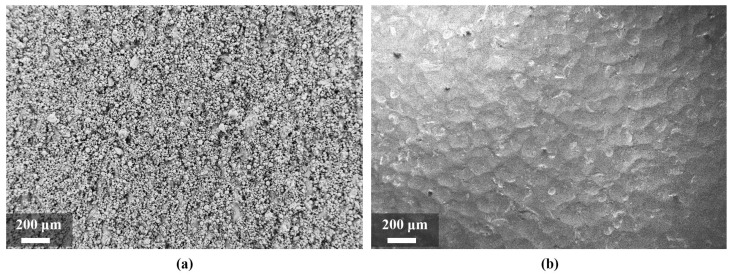
FESEM images of the surface roughness on the lateral surface of the sample 6 before (**a**) and after (**b**) shot blasting with glass beads.

**Figure 9 materials-15-03352-f009:**
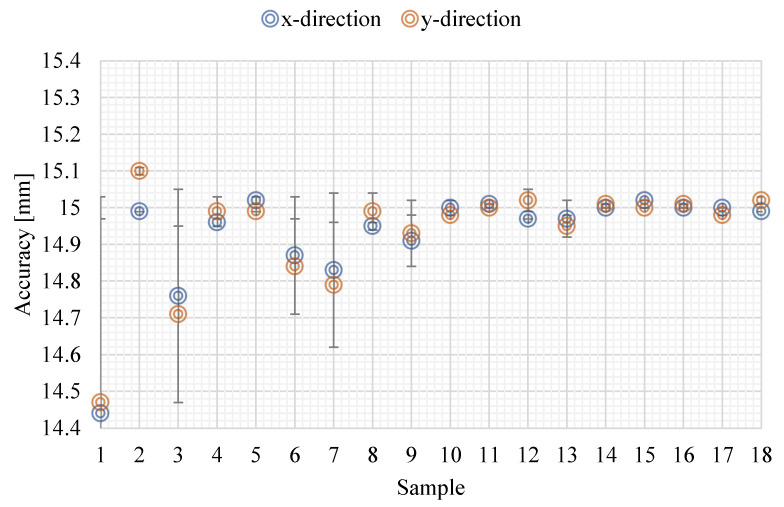
Accuracy in x- and y-directions.

**Figure 10 materials-15-03352-f010:**
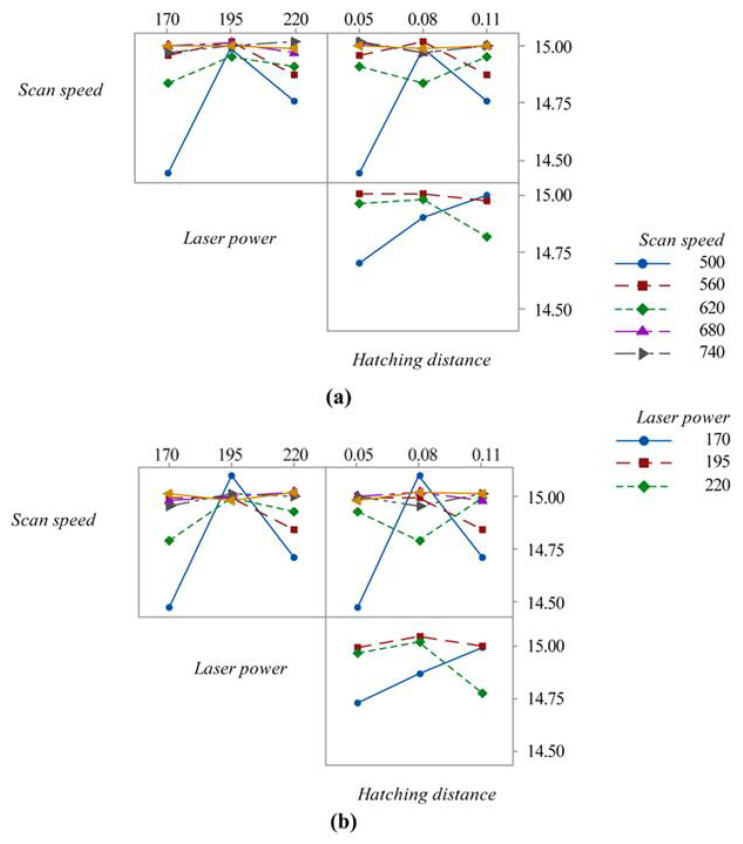
Interaction plot between process parameters and accuracy: (**a**) x-direction, (**b**) y-direction.

**Table 1 materials-15-03352-t001:** Measured and declared chemical compositions of Cu-10Sn alloy.

Element	Cu	Sn	P	Bi	Al	Si	Cr	Co	Fe	Other
Content measured (wt.%)	88.7	10.9	0.351	0.013	0.007	0.006	0.006	0.005	0.004	0.008
Content declared (wt%)	Bal.	10.92	0.33	-	-	-	-	-	-	-

**Table 2 materials-15-03352-t002:** Variable and fixed process parameters employed.

Variable		Fixed	
Parameters	Values	Parameters	Values
*v* (mm/s)	500, 560, 620, 680, 740, 800	Layer thickness (µm)	30
*P* (W)	170, 195, 220	Spot size (mm)	0.1
*h_d_* (mm)	0.05, 0.08, 0.11	*v_contour_* (mm/s)	900
		*P_contour_* (W)	100

**Table 3 materials-15-03352-t003:** Process parameters, density and hardness results.

Sample	Process Parameters			Volume Rate [cm^3^/h]	Energy Density [J/mm^3^]	Experimental Density [%]	Hardness [HV]
	*v* [mm/s]	*P* [W]	*h_d_* [mm]				
1	500	170	0.05	2.70	226.7	99.48 ± 0.15	162 ± 15
2	500	195	0.08	4.32	162.5	99.81 ± 0.15	162 ± 6
3	500	220	0.11	5.94	133.3	99.86 ± 0.06	157 ± 7
4	560	170	0.05	3.02	202.4	99.6 ± 0.2	162 ± 10
5	560	195	0.08	4.84	145.1	99.84 ± 0.13	163 ± 4
6	560	220	0.11	6.65	119.0	99.8 ± 0.1	162 ± 13
7	620	170	0.08	5.36	114.2	99.4 ± 0.3	166 ± 4
8	620	195	0.11	7.37	95.3	99.47 ± 0.24	154 ± 7
9	620	220	0.05	3.35	236.6	99.7 ± 0.1	165 ± 4
10	680	170	0.11	8.08	75.8	99.32 ± 0.18	142 ± 13
11	680	195	0.05	3.67	191.2	99.52 ± 0.19	153 ± 7
12	680	220	0.08	5.88	134.8	99.66 ± 0.12	159 ± 12
13	740	170	0.08	6.39	95.7	99.33 ± 0.16	160 ± 3
14	740	195	0.11	8.79	79.9	99.5 ± 0.4	158 ± 1
15	740	220	0.05	4.00	198.2	99.83 ± 0.13	152 ± 12
16	800	170	0.11	9.50	64.4	98.7 ± 0.4	163 ± 2
17	800	195	0.05	4.32	162.5	99.64 ± 0.12	153 ± 6
18	800	220	0.08	6.91	114.6	99.77 ± 0.06	150 ± 15

**Table 4 materials-15-03352-t004:** Average S/N ratios.

Level	Larger-the-Better–S/N (dB)		
	Scan Speed	Laser Power	Hatching Distance
1	39.98	39.94	39.97
2	39.98	39.97	39.97
3	39.96	39.98	39.95
4	39.96		
5	39.96		
6	39.95		
Delta	0.03	0.04	0.02
Rank	2	1	3

**Table 5 materials-15-03352-t005:** Analysis of Variance for S/N ratios. *R*^2^ = 85.26%.

Source	DOF	Sum of Squares	F	*p*	Statistical Significance
Scan speed	5	19.875	2.39	0.131	Not significant
Laser power	2	48.176	24.46	0.002	Highly Significant
Hatching distance	2	9.026	2.71	0.126	Not Significant
Residual Error	8	13.323			
Total	17	90.400			

**Table 6 materials-15-03352-t006:** Average S/N ratios.

Level	Smaller-the-Better–S/N (dB)		
	Scan Speed	Laser Power	Hatching Distance
1	−14.84	−19.22	−16.83
2	−15.04	−16.61	−16.01
3	−17.49	−14.44	−17.44
4	−18.10		
5	−17.37		
6	−17.71		
Delta	3.25	4.78	1.43
Rank	2	1	3

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
