# Peer review of "Production of Dense Cu-10Sn Part by Laser Powder Bed Fusion with Low Surface Roughness and High Dimensional Accuracy"

_materials, 2022, doi:10.3390/ma15093352_

Round 1

Reviewer 1 Report

The paper is totally fine. 

For the density measurement, maybe it is better to use Gas Pycnometer (Helium True Density) for comparison.

In the introduction part, it is better to cite some more recent papers about the preparation of novel metallic composites by powder metallurgy, e.g., Mater. Des. 2021, 210: 110108.

Author Response

We highly appreciate the comments of the Reviewer 1 and deeply thank him/her for the valuable suggestions on our manuscript. We have answered to all the comments and modified the manuscript accordingly, indicating all changes made in the text.

Reviewer 2 Report

The current manuscript investigates the effect of LPBF process parameters on the density, surface roughness, and dimensional accuracy of Cu-10Sn parts. The manuscript is well organized and there are some interesting results presented. However, some issues should be considered as following: 

  • The title should be adjusted to reflect the presented data in the manuscript including the effect of surface roughness and dimensional accuracy as well as density. 
  • The introduction should present a critical review to show the effect of the LPBF process parameters on the quality of additively manufactured parts for different materials. Some comprehensive studies can be cited such as the following:

 Maamoun, A.H.; Xue, Y.F.; Elbestawi, M.A.; Veldhuis, S.C. Effect of Selective Laser Melting Process Parameters on the Quality of Al Alloy Parts: Powder Characterization, Density, Surface Roughness, and Dimensional Accuracy. Materials 2018, 11, 2343. https://doi.org/10.3390/ma11122343 

Singla, Anil Kumar, et al. "Selective laser melting of Ti6Al4V alloy: Process parameters, defects and post-treatments." Journal of Manufacturing Processes 64 (2021): 161-187.

Maamoun, A.H.; Xue, Y.F.; Elbestawi, M.A.; Veldhuis, S.C. The Effect of Selective Laser Melting Process Parameters on the Microstructure and Mechanical Properties of Al6061 and AlSi10Mg Alloys. Materials 2019, 12, 12. https://doi.org/10.3390/ma12010012

  • Line #62, the value of layer thickness is missing.
  • ‏The particle size distribution should be added, which standard was applied for the powder characterization test?
  • The authors stated that the build platform is preheated to 80C, what is the reference for choosing this option? and what is the effect of platform preheating? that should be included.
  • Please cite the references that help in selecting the range of process parameters values.
  •  Table 3, the results of sample #3 is relatively lower than other values, please illustrate that in the discussion. 
  • Figure 3; please label the title of vertical axis.
  • The standard deviation values should be added to Figure 4 as reported in Table 3.
  • The data presented in Figure 5 should be re-arranged to clearly display the measured surface roughness for the samples.
  • Figure 7 should include images for the samples after shot blasting.
  • The standard deviation should be added to the values in Figure 8. 
  • In general, the discussion should be supported with more references to justify the presented results.

Author Response

We highly appreciate the comments of the Reviewer 2 and deeply thank him/her for the valuable suggestions on our manuscript. We have answered to all the comments and modified the manuscript accordingly, indicating all changes made in the text.

Reviewer 3 Report

In this research work, a set of SLM-process parameters was identified to produce fully dense Cu-10Sn samples with low surface roughness and high accuracy. Given the intensive development of additive technologies, the article is relevant. However, for publication in high-ranking scientific journals, a moderate revision is required. The questions are addressed as follow:

1. The significance of the work is not clear. In the Introduction Section, the authors refer to numerous works on the SLM-preparation of Cu-10Sn alloys, but do not point out the shortcomings of these studies compared to their own. Except, perhaps, Ref. [11], where there is a slight microcracking.

2. Lines 193-194. ‘…while with Archimedes' method, values higher than the theoretical density value were obtained’. But in Table 3, the density of all samples is less than 100%, i.e. less than theoretical. Clarify please.

Other:

- Line 186. Should be ‘99.6 ± 0.3%’ instead ‘99.6% ± 0.3’

- Line 232. Should be ‘163 ± 2 HV’ instead ‘163 HV ± 2’

Author Response

We highly appreciate the comments of the Reviewer 3 and deeply thank him/her for the valuable suggestions on our manuscript. We have answered to all the comments and modified the manuscript accordingly, indicating all changes made in the text.

Round 2

Reviewer 1 Report

Publish it. Good luck!

Reviewer 2 Report

The revised manuscript is significantly improved. All review comments and recommendations are well addressed.

Reviewer 3 Report

The paper was improved, it can be accepted as it is.